# DECOUPLED WEIGHT DECAY REGULARIZATION

**Ilya Loshchilov & Frank Hutter**
University of Freiburg
Freiburg, Germany,
`ilya.loshchilov@gmail.com, fh@cs.uni-freiburg.de`

## ABSTRACT

$L_2$ regularization and weight decay regularization are equivalent for standard stochastic gradient descent (when rescaled by the learning rate), but as we demonstrate this is *not* the case for adaptive gradient algorithms, such as Adam. While common implementations of these algorithms employ $L_2$ regularization (often calling it "weight decay" in what may be misleading due to the inequivalence we expose), we propose a simple modification to recover the original formulation of weight decay regularization by *decoupling* the weight decay from the optimization steps taken w.r.t. the loss function. We provide empirical evidence that our proposed modification (i) decouples the optimal choice of weight decay factor from the setting of the learning rate for both standard SGD and Adam and (ii) substantially improves Adam's generalization performance, allowing it to compete with SGD with momentum on image classification datasets (on which it was previously typically outperformed by the latter). Our proposed decoupled weight decay has already been adopted by many researchers, and the community has implemented it in TensorFlow and PyTorch; the complete source code for our experiments is available at `https://github.com/loshchil/AdamW-and-SGDW`

## 1 INTRODUCTION

Adaptive gradient methods, such as AdaGrad (Duchi et al., 2011), RMSProp (Tieleman & Hinton, 2012), Adam (Kingma & Ba, 2014) and most recently AMSGrad (Reddi et al., 2018) have become a default method of choice for training feed-forward and recurrent neural networks (Xu et al., 2015; Radford et al., 2015). Nevertheless, state-of-the-art results for popular image classification datasets, such as CIFAR-10 and CIFAR-100 Krizhevsky (2009), are still obtained by applying SGD with momentum (Gastaldi, 2017; Cubuk et al., 2018). Furthermore, Wilson et al. (2017) suggested that adaptive gradient methods do not generalize as well as SGD with momentum when tested on a diverse set of deep learning tasks, such as image classification, character-level language modeling and constituency parsing. Different hypotheses about the origins of this worse generalization have been investigated, such as the presence of sharp local minima (Keskar et al., 2016; Dinh et al., 2017) and inherent problems of adaptive gradient methods (Wilson et al., 2017). In this paper, we investigate whether it is better to use $L_2$ regularization or weight decay regularization to train deep neural networks with SGD and Adam. We show that a major factor of the poor generalization of the most popular adaptive gradient method, Adam, is due to the fact that $L_2$ regularization is not nearly as effective for it as for SGD. Specifically, our analysis of Adam leads to the following observations:

**$L_2$ regularization and weight decay are not identical.** Contrary to a belief which seems popular among some practitioners, the two techniques are not equivalent. For SGD, they can be made equivalent by a reparameterization of the weight decay factor based on the learning rate; this is not the case for Adam. In particular, when combined with adaptive gradients, $L_2$ regularization leads to weights with large parameter and/or gradient amplitudes being regularized less than they would be when using weight decay.

**$L_2$ regularization is not effective in Adam.** One possible explanation why Adam and other adaptive gradient methods might be outperformed by SGD with momentum is that common deep learning libraries only implement $L_2$ regularization, not the original weight decay. Therefore, on tasks/datasets where the use of $L_2$ regularization is beneficial for SGD (e.g.,

on many popular image classification datasets), Adam leads to worse results than SGD with momentum (for which $L_2$ regularization behaves as expected).

**Weight decay is equally effective in both SGD and Adam.** For SGD, it is equivalent to $L_2$ regularization, while for Adam it is not.

**Optimal weight decay depends on the total number of batch passes/weight updates.** Our empirical analysis of SGD and Adam suggests that the larger the runtime/number of batch passes to be performed, the smaller the optimal weight decay. This effect tends to be neglected because hyperparameters are often tuned for a fixed number of training epochs. As a result, the values of the weight decay found to perform best for short runs do not generalize to much longer runs.

The main contribution of this paper is to *improve regularization in Adam by decoupling the weight decay from the gradient-based update*. In a comprehensive analysis, we show that Adam generalizes substantially better with decoupled weight decay than with $L_2$ regularization, achieving 15% relative improvement in test error (see Figures 2 and 3); this holds true for various image recognition datasets (CIFAR-10 and ImageNet32x32), training budgets (ranging from 100 to 1800 epochs), and learning rate schedules (fixed, drop-step, and cosine annealing; see Figure 1). We demonstrate that our decoupled weight decay renders the optimal settings of the learning rate and the weight decay factor much more independent, thereby easing hyperparameter optimization (see Figure 2).

The main motivation of this paper is to improve Adam to make it competitive w.r.t. SGD with momentum even for those problems where it did not use to be competitive. We hope that as a result, practitioners do not need to switch between Adam and SGD anymore, which in turn should reduce the common issue of selecting dataset/task-specific training algorithms and their hyperparameters.

## 2  DECOUPLING THE WEIGHT DECAY FROM THE GRADIENT-BASED UPDATE

In the weight decay described by Hanson & Pratt (1988), the weights $\boldsymbol{\theta}$ decay exponentially as

$$\boldsymbol{\theta}_{t+1} = (1 - \lambda)\boldsymbol{\theta}_t - \alpha \nabla f_t(\boldsymbol{\theta}_t), \tag{1}$$

where $\lambda$ defines the rate of the weight decay per step and $\nabla f_t(\boldsymbol{\theta}_t)$ is the $t$-th batch gradient to be multiplied by a learning rate $\alpha$. For standard SGD, it is equivalent to standard $L_2$ regularization:

**Proposition 1** (Weight decay = $L_2$ reg for standard SGD). *Standard SGD with base learning rate $\alpha$ executes the same steps on batch loss functions $f_t(\boldsymbol{\theta})$ with weight decay $\lambda$ (defined in Equation 1) as it executes without weight decay on $f_t^{reg}(\boldsymbol{\theta}) = f_t(\boldsymbol{\theta}) + \frac{\lambda'}{2} \|\boldsymbol{\theta}\|_2^2$, with $\lambda' = \frac{\lambda}{\alpha}$.*

The proofs of this well-known fact, as well as our other propositions, are given in the Appendix A.

Due to this equivalence, $L_2$ regularization is very frequently referred to as weight decay, including in popular deep learning libraries. However, as we will demonstrate later in this section, this equivalence does *not* hold for adaptive gradient methods. One fact that is often overlooked already for the simple case of SGD is that in order for the equivalence to hold, the $L_2$ regularizer $\lambda'$ has to be set to $\frac{\lambda}{\alpha}$, i.e., if there is an overall best weight decay value $\lambda$, the best value of $\lambda'$ is tightly coupled with the learning rate $\alpha$. In order to decouple the effects of these two hyperparameters, we advocate to decouple the weight decay step as proposed by Hanson & Pratt (1988) (Equation 1).

Looking first at the case of SGD, we propose to decay the weights simultaneously with the update of $\boldsymbol{\theta}_t$ based on gradient information in Line 9 of Algorithm 1. This yields our proposed variant of SGD with momentum using decoupled weight decay (**SGDW**). This simple modification explicitly decouples $\lambda$ and $\alpha$ (although some problem-dependent implicit coupling may of course remain as for any two hyperparameters). In order to account for a possible scheduling of both $\alpha$ and $\lambda$, we introduce a scaling factor $\eta_t$ delivered by a user-defined procedure $SetScheduleMultiplier(t)$.

Now, let's turn to adaptive gradient algorithms like the popular optimizer Adam Kingma & Ba (2014), which scale gradients by their historic magnitudes. Intuitively, when Adam is run on a loss function $f$ plus $L_2$ regularization, weights that tend to have large gradients in $f$ do not get regularized as much as they would with decoupled weight decay, since the gradient of the regularizer gets scaled along with the gradient of $f$. This leads to an inequivalence of $L_2$ and decoupled weight decay regularization for adaptive gradient algorithms:

**Algorithm 1** SGD with L$_2$ regularization and SGD with decoupled weight decay (SGDW), both with momentum

1: **given** initial learning rate $\alpha \in \mathbb{R}$, momentum factor $\beta_1 \in \mathbb{R}$, weight decay/L$_2$ regularization factor $\lambda \in \mathbb{R}$

2: **initialize** time step $t \leftarrow 0$, parameter vector $\boldsymbol{\theta}_{t=0} \in \mathbb{R}^n$, first moment vector $\boldsymbol{m}_{t=0} \leftarrow \boldsymbol{0}$, schedule multiplier $\eta_{t=0} \in \mathbb{R}$
3: **repeat**
4:     $t \leftarrow t + 1$
5:     $\nabla f_t(\boldsymbol{\theta}_{t-1}) \leftarrow \text{SelectBatch}(\boldsymbol{\theta}_{t-1})$                 ▷ select batch and return the corresponding gradient
6:     $\boldsymbol{g}_t \leftarrow \nabla f_t(\boldsymbol{\theta}_{t-1}) \;\boxed{+\lambda\boldsymbol{\theta}_{t-1}}$
7:     $\eta_t \leftarrow \text{SetScheduleMultiplier}(t)$                     ▷ can be fixed, decay, be used for warm restarts
8:     $\boldsymbol{m}_t \leftarrow \beta_1\boldsymbol{m}_{t-1} + \eta_t\alpha\boldsymbol{g}_t$
9:     $\boldsymbol{\theta}_t \leftarrow \boldsymbol{\theta}_{t-1} - \boldsymbol{m}_t \;\boxed{-\eta_t\lambda\boldsymbol{\theta}_{t-1}}$
10: **until** *stopping criterion is met*
11: **return** optimized parameters $\boldsymbol{\theta}_t$

---

**Algorithm 2** Adam with L$_2$ regularization and Adam with decoupled weight decay (AdamW)

1: **given** $\alpha = 0.001, \beta_1 = 0.9, \beta_2 = 0.999, \epsilon = 10^{-8}, \lambda \in \mathbb{R}$
2: **initialize** time step $t \leftarrow 0$, parameter vector $\boldsymbol{\theta}_{t=0} \in \mathbb{R}^n$, first moment vector $\boldsymbol{m}_{t=0} \leftarrow \boldsymbol{0}$, second moment vector $\boldsymbol{v}_{t=0} \leftarrow \boldsymbol{0}$, schedule multiplier $\eta_{t=0} \in \mathbb{R}$
3: **repeat**
4:     $t \leftarrow t + 1$
5:     $\nabla f_t(\boldsymbol{\theta}_{t-1}) \leftarrow \text{SelectBatch}(\boldsymbol{\theta}_{t-1})$                 ▷ select batch and return the corresponding gradient
6:     $\boldsymbol{g}_t \leftarrow \nabla f_t(\boldsymbol{\theta}_{t-1}) \;\boxed{+\lambda\boldsymbol{\theta}_{t-1}}$
7:     $\boldsymbol{m}_t \leftarrow \beta_1\boldsymbol{m}_{t-1} + (1 - \beta_1)\boldsymbol{g}_t$                     ▷ here and below all operations are element-wise
8:     $\boldsymbol{v}_t \leftarrow \beta_2\boldsymbol{v}_{t-1} + (1 - \beta_2)\boldsymbol{g}_t^2$
9:     $\hat{\boldsymbol{m}}_t \leftarrow \boldsymbol{m}_t/(1 - \beta_1^t)$                         ▷ $\beta_1$ is taken to the power of $t$
10:     $\hat{\boldsymbol{v}}_t \leftarrow \boldsymbol{v}_t/(1 - \beta_2^t)$                         ▷ $\beta_2$ is taken to the power of $t$
11:     $\eta_t \leftarrow \text{SetScheduleMultiplier}(t)$                 ▷ can be fixed, decay, or also be used for warm restarts
12:     $\boldsymbol{\theta}_t \leftarrow \boldsymbol{\theta}_{t-1} - \eta_t\left(\alpha\hat{\boldsymbol{m}}_t/(\sqrt{\hat{\boldsymbol{v}}_t} + \epsilon) \;\boxed{+\lambda\boldsymbol{\theta}_{t-1}}\right)$
13: **until** *stopping criterion is met*
14: **return** optimized parameters $\boldsymbol{\theta}_t$

---

**Proposition 2** (Weight decay $\neq$ L$_2$ reg for adaptive gradients). *Let $O$ denote an optimizer that has iterates $\boldsymbol{\theta}_{t+1} \leftarrow \boldsymbol{\theta}_t - \alpha\mathbf{M}_t\nabla f_t(\boldsymbol{\theta}_t)$ when run on batch loss function $f_t(\boldsymbol{\theta})$ without weight decay, and $\boldsymbol{\theta}_{t+1} \leftarrow (1 - \lambda)\boldsymbol{\theta}_t - \alpha\mathbf{M}_t\nabla f_t(\boldsymbol{\theta}_t)$ when run on $f_t(\boldsymbol{\theta})$ with weight decay, respectively, with $\mathbf{M}_t \neq k\mathbf{I}$ (where $k \in \mathbb{R}$). Then, for $O$ there exists no L$_2$ coefficient $\lambda'$ such that running $O$ on batch loss $f_t^{reg}(\boldsymbol{\theta}) = f_t(\boldsymbol{\theta}) + \frac{\lambda'}{2}\|\boldsymbol{\theta}\|_2^2$ without weight decay is equivalent to running $O$ on $f_t(\boldsymbol{\theta})$ with decay $\lambda \in \mathbb{R}^+$.*

We decouple weight decay and loss-based gradient updates in Adam as shown in line 12 of Algorithm 2; this gives rise to our variant of Adam with decoupled weight decay (**AdamW**).

Having shown that L$_2$ regularization and weight decay regularization differ for adaptive gradient algorithms raises the question of how they differ and how to interpret their effects. Their equivalence for standard SGD remains very helpful for intuition: both mechanisms push weights closer to zero, at the same rate. However, for adaptive gradient algorithms they differ: with L$_2$ regularization, the sums of the gradient of the loss function and the gradient of the regularizer (i.e., the L$_2$ norm of the weights) are adapted, whereas with weight decay, only the gradients of the loss function are adapted (with the weight decay step separated from the adaptive gradient mechanism). With L$_2$ regularization both types of gradients are normalized by their typical (summed) magnitudes, and therefore weights $x$ with large typical gradient magnitude $s$ are regularized by a smaller relative amount than other weights. In contrast, weight decay regularizes all weights with the same rate $\lambda$, effectively regularizing weights $x$ with large $s$ more than standard L$_2$ regularization does. We demonstrate this formally for a simple special case of adaptive gradient algorithm with a fixed preconditioner:

**Proposition 3** (Weight decay = scale-adjusted $L_2$ reg for adaptive gradient algorithm with fixed preconditioner). *Let $O$ denote an algorithm with the same characteristics as in Proposition 2, and using a fixed preconditioner matrix $\boldsymbol{M}_t = diag(\boldsymbol{s})^{-1}$ (with $s_i > 0$ for all $i$). Then, $O$ with base learning rate $\alpha$ executes the same steps on batch loss functions $f_t(\boldsymbol{\theta})$ with weight decay $\lambda$ as it executes without weight decay on the scale-adjusted regularized batch loss*

$$f_t^{sreg}(\boldsymbol{\theta}) = f_t(\boldsymbol{\theta}) + \frac{\lambda'}{2\alpha} \left\| \boldsymbol{\theta} \odot \sqrt{\boldsymbol{s}} \right\|_2^2, \tag{2}$$

*where $\odot$ and $\sqrt{\cdot}$ denote element-wise multiplication and square root, respectively, and $\lambda' = \frac{\lambda}{\alpha}$.*

# 3 Justification of Decoupled Weight Decay via a View of Adaptive Gradient Methods as Bayesian Filtering

We now discuss a justification of decoupled weight decay in the framework of Bayesian filtering for a unified theory of adaptive gradient algorithms due to Aitchison (2018). After we posted a preliminary version of our current paper on arXiv, Aitchison noted that his theory "gives us a theoretical framework in which we can understand the superiority of this weight decay over $L_2$ regularization, because it is weight decay, rather than $L_2$ regularization that emerges through the straightforward application of Bayesian filtering."(Aitchison, 2018). While full credit for this theory goes to Aitchison, we summarize it here to shed some light on why weight decay may be favored over $L_2$ regularization.

Aitchison (2018) views stochastic optimization of $n$ parameters $x_1, \ldots, x_n$ as a Bayesian filtering problem with the goal of inferring a distribution over the optimal values of each of the parameters $x_i$ given the current values of the other parameters $\boldsymbol{\theta}_{-i}(t)$ at time step $t$. When the other parameters do not change this is an optimization problem, but when they do change it becomes one of "tracking" the optimizer using Bayesian filtering as follows. One is given a probability distribution $P(\boldsymbol{\theta}_t \mid \boldsymbol{y}_{1:t})$ of the optimizer at time step $t$ that takes into account the data $\boldsymbol{y}_{1:t}$ from the first $t$ mini batches, a state transition prior $P(\boldsymbol{\theta}_{t+1} \mid \boldsymbol{\theta}_t)$ reflecting a (small) data-independent change in this distribution from one step to the next, and a likelihood $P(\boldsymbol{y}_{t+1} \mid \boldsymbol{\theta}_{t+1})$ derived from the mini batch at step $t + 1$. The posterior distribution $P(\boldsymbol{\theta}_{t+1} \mid \boldsymbol{y}_{1:t+1})$ of the optimizer at time step $t + 1$ can then be computed (as usual in Bayesian filtering) by marginalizing over $\boldsymbol{\theta}_t$ to obtain the one-step ahead predictions $P(\boldsymbol{\theta}_{t+1} \mid \boldsymbol{y}_{1:t})$ and then applying Bayes' rule to incorporate the likelihood $P(\boldsymbol{y}_{t+1} \mid \boldsymbol{\theta}_t)$. Aitchison (2018) assumes a Gaussian state transition distribution $P(\boldsymbol{\theta}_{t+1} \mid \boldsymbol{\theta}_t)$ and an approximate conjugate likelihood $P(\boldsymbol{y}_{t+1} \mid \boldsymbol{\theta}_{t+1})$, leading to the following closed-form update of the filtering distribution's mean:

$$\boldsymbol{\mu}_{post} = \boldsymbol{\mu}_{prior} + \boldsymbol{\Sigma}_{post} \times \boldsymbol{g}, \tag{3}$$

where $\boldsymbol{g}$ is the gradient of the log likelihood of the mini batch at time $t$. This result implies a preconditioner of the gradients that is given by the posterior uncertainty $\boldsymbol{\Sigma}_{post}$ of the filtering distribution: updates are larger for parameters we are more uncertain about and smaller for parameters we are more certain about. Aitchison (2018) goes on to show that popular adaptive gradient methods, such as Adam and RMSprop, as well as Kronecker-factorized methods are special cases of this framework.

Decoupled weight decay very naturally fits into this unified framework can express weight decay as part of the state-transition distribution: Aitchison (2018) assumes a slow change of the optimizer according to the following Gaussian:

$$P(\boldsymbol{\theta}_{t+1} \mid \boldsymbol{\theta}_t) = \mathcal{N}((\boldsymbol{I} - \boldsymbol{A})\boldsymbol{\theta}_t, \boldsymbol{Q}), \tag{4}$$

where $\boldsymbol{Q}$ is the covariance of Gaussian perturbations of the weights, and $\boldsymbol{A}$ is a regularizer to avoid values growing unboundedly over time. When instantiated as $\boldsymbol{A} = \lambda \times \boldsymbol{I}$, this regularizer $\boldsymbol{A}$ plays exactly the role of decoupled weight decay as described in Equation 1, since this leads to multiplying the current mean estimate $\boldsymbol{\theta}_t$ by $(1 - \lambda)$ at each step. Notably, this regularization is also directly applied to the prior and does not depend on the uncertainty in each of the parameters (which would be required for $L_2$ regularization).

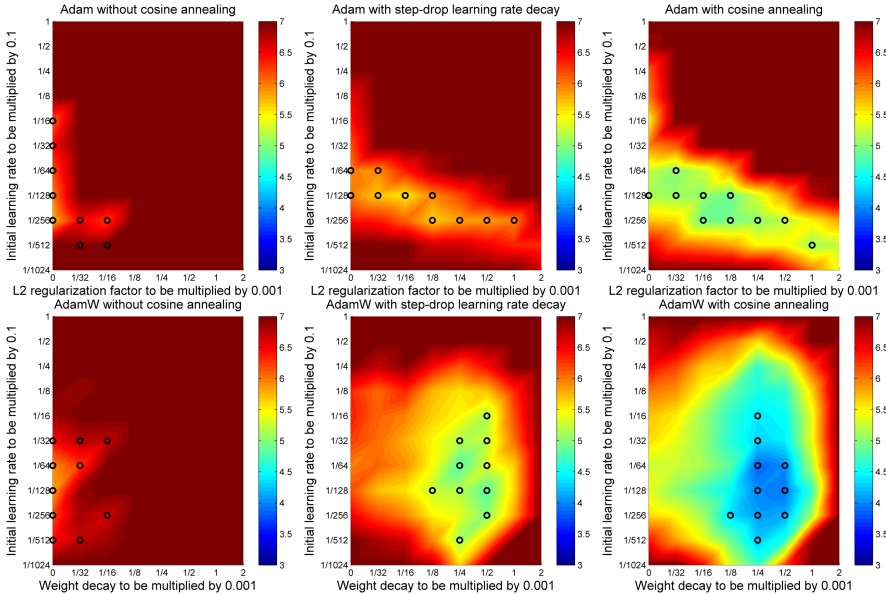

Figure 1: Adam performs better with decoupled weight decay (bottom row, AdamW) than with $L_2$ regularization (top row, Adam). We show the final test error of a 26 2x64d ResNet on CIFAR-10 after 100 epochs of training with fixed learning rate (left column), step-drop learning rate (with drops at epoch indexes 30, 60 and 80, middle column) and cosine annealing (right column). AdamW leads to a more separable hyperparameter search space, especially when a learning rate schedule, such as step-drop and cosine annealing is applied. Cosine annealing yields clearly superior results.

## 4 EXPERIMENTAL VALIDATION

We now evaluate the performance of decoupled weight decay under various training budgets and learning rate schedules. Our experimental setup follows that of Gastaldi (2017), who proposed, in addition to L$_2$ regularization, to apply the new Shake-Shake regularization to a 3-branch residual DNN that allowed to achieve new state-of-the-art results of 2.86% on the CIFAR-10 dataset (Krizhevsky, 2009). We always used a batch size of 128. The regular data augmentation procedure used for the CIFAR datasets was applied. We used the same model/source code based on fb.resnet.torch [1]. The base networks are a 26 2x64d ResNet (i.e. the network has a depth of 26, 2 residual branches and the first residual block has a width of 64) and a 26 2x96d ResNet with 11.6M and 25.6M parameters, respectively. For a detailed description of the network and the Shake-Shake method, we refer the interested reader to Gastaldi (2017). We also perform experiments on the ImageNet32x32 dataset (Chrabaszcz et al., 2017), a downsampled version of the original ImageNet dataset with 1.2 million 32×32 pixels images.

### 4.1 EVALUATING DECOUPLED WEIGHT DECAY WITH DIFFERENT LEARNING RATE SCHEDULES

In our first experiment, we compare Adam with $L_2$ regularization to Adam with decoupled weight decay (AdamW), using three different learning rate schedules: a fixed learning rate, a drop-step schedule, and a cosine annealing schedule (Loshchilov & Hutter, 2016). For each learning rate schedule and weight decay variant, we trained a 2x64d ResNet for 100 epochs, using different settings of the initial learning rate $\alpha$ and the weight decay factor $\lambda$. Figure 1 shows that decoupled weight decay outperforms $L_2$ regularization for all learning rate schedules, with larger differences for better learning rate schedules. We also note that decoupled weight decay leads to a more separable hyperparameter search space, especially when a learning rate schedule, such as step-drop and

---

[1]https://github.com/xgastaldi/shake-shake

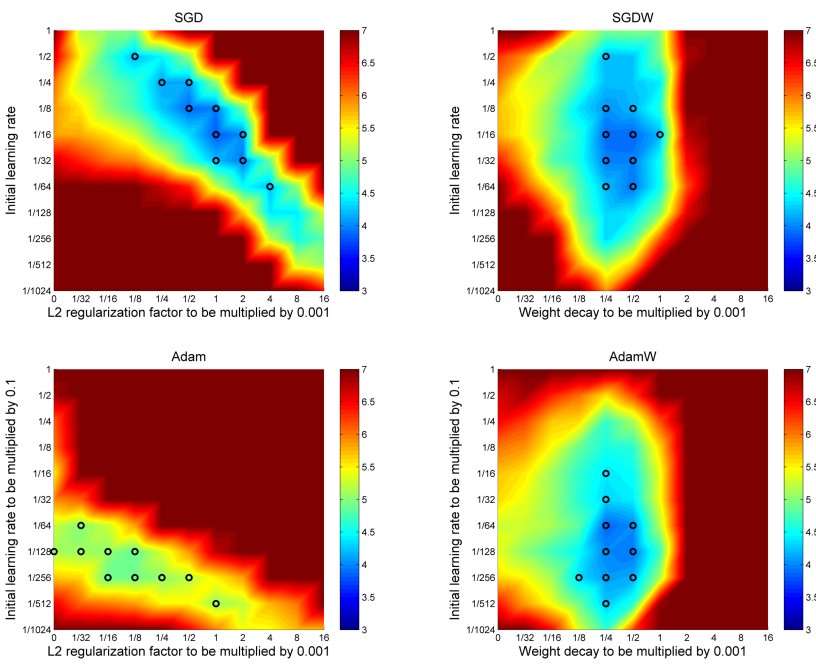

Figure 2: The Top-1 test error of a 26 2x64d ResNet on CIFAR-10 measured after 100 epochs. The proposed SGDW and AdamW (right column) have a more separable hyperparameter space.

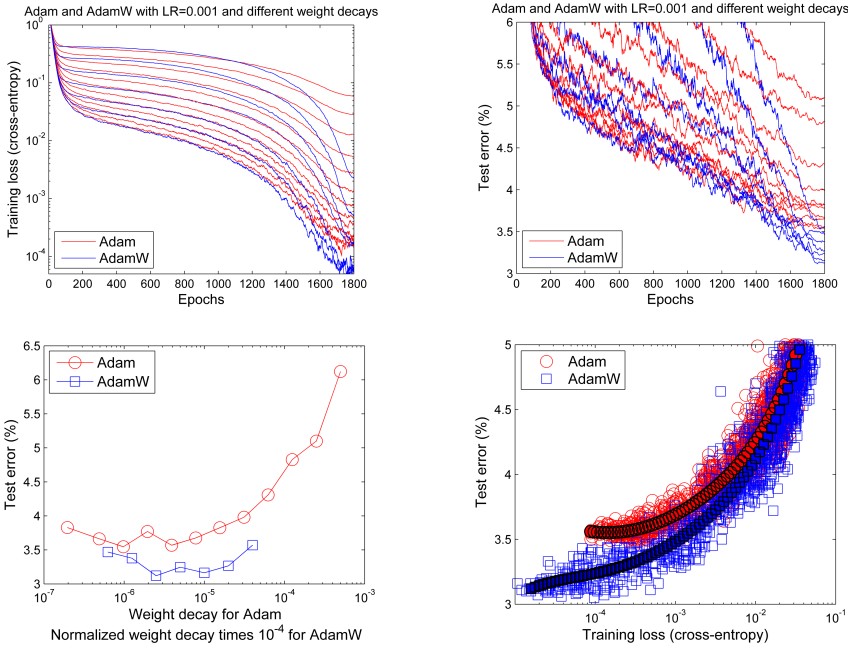

Figure 3: Learning curves (top row) and generalization results (bottom row) obtained by a 26 2x96d ResNet trained with Adam and AdamW on CIFAR-10. See text for details. SuppFigure 4 in the Appendix shows the same qualitative results for ImageNet32x32.

cosine annealing is applied. The figure also shows that cosine annealing clearly outperforms the other learning rate schedules; we thus used cosine annealing for the remainder of the experiments.

## 4.2 Decoupling the Weight Decay and Initial Learning Rate Parameters

In order to verify our hypothesis about the coupling of $\alpha$ and $\lambda$, in Figure 2 we compare the performance of $L_2$ regularization vs. decoupled weight decay in SGD (SGD vs. SGDW, top row) and in Adam (Adam vs. AdamW, bottom row). In SGD (Figure 2, top left), $L_2$ regularization is not decoupled from the learning rate (the common way as described in Algorithm 1), and the figure clearly shows that the basin of best hyperparameter settings (depicted by color and top-10 hyperparameter settings by black circles) is not aligned with the x-axis or y-axis but lies on the diagonal. This suggests that the two hyperparameters are interdependent and need to be changed simultaneously, while only changing one of them might substantially worsen results. Consider, e.g., the setting at the top left black circle ($\alpha = 1/2$, $\lambda = 1/8 * 0.001$); only changing either $\alpha$ or $\lambda$ by itself would worsen results, while changing both of them could still yield clear improvements. We note that this coupling of initial learning rate and $L_2$ regularization factor might have contributed to SGD's reputation of being very sensitive to its hyperparameter settings.

In contrast, the results for SGD with decoupled weight decay (SGDW) in Figure 2 (top right) show that weight decay and initial learning rate are decoupled. The proposed approach renders the two hyperparameters more separable: even if the learning rate is not well tuned yet (e.g., consider the value of 1/1024 in Figure 2, top right), leaving it fixed and only optimizing the weight decay factor would yield a good value (of 1/4*0.001). This is not the case for SGD with $L_2$ regularization (see Figure 2, top left).

The results for Adam with $L_2$ regularization are given in Figure 2 (bottom left). Adam's best hyperparameter settings performed clearly worse than SGD's best ones (compare Figure 2, top left). While both methods used $L_2$ regularization, Adam did not benefit from it at all: its best results obtained for non-zero $L_2$ regularization factors were comparable to the best ones obtained without the $L_2$ regularization, i.e., when $\lambda = 0$. Similarly to the original SGD, the shape of the hyperparameter landscape suggests that the two hyperparameters are coupled.

In contrast, the results for our new variant of Adam with decoupled weight decay (AdamW) in Figure 2 (bottom right) show that AdamW largely decouples weight decay and learning rate. The results for the best hyperparameter settings were substantially better than the best ones of Adam with $L_2$ regularization and rivaled those of SGD and SGDW.

In summary, the results in Figure 2 support our hypothesis that the weight decay and learning rate hyperparameters can be decoupled, and that this in turn simplifies the problem of hyperparameter tuning in SGD and improves Adam's performance to be competitive w.r.t. SGD with momentum.

## 4.3 Better Generalization of AdamW

While the previous experiment suggested that the basin of optimal hyperparameters of AdamW is broader and deeper than the one of Adam, we next investigated the results for much longer runs of 1800 epochs to compare the generalization capabilities of AdamW and Adam.

We fixed the initial learning rate to 0.001 which represents both the default learning rate for Adam and the one which showed reasonably good results in our experiments. Figure 3 shows the results for 12 settings of the $L_2$ regularization of Adam and 7 settings of the normalized weight decay of AdamW (the normalized weight decay represents a rescaling formally defined in the Appendix B.1, it amounts to a multiplicative factor which depends on the number of bath passes). Interestingly, while the dynamics of the learning curves of Adam and AdamW often coincided for the first half of the training run, AdamW often led to lower training loss and test errors (see Figure 3 top left and top right, respectively). Importantly, the use of weight decay in Adam did not yield as good results as in AdamW (see also Figure 3, bottom left). Next, we investigated whether AdamW's better results were only due to better convergence or due to better generalization. *The results in Figure 3 (bottom right) for the best settings of Adam and AdamW suggest that AdamW did not only yield better training loss but also yielded better generalization performance for similar training loss values.* The results on ImageNet32x32 (see SuppFigure 4 in the Appendix) lead to the same conclusion of substantially improved generalization performance.

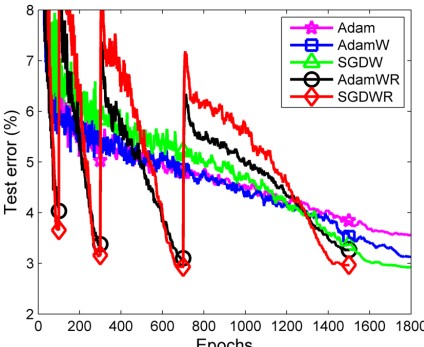 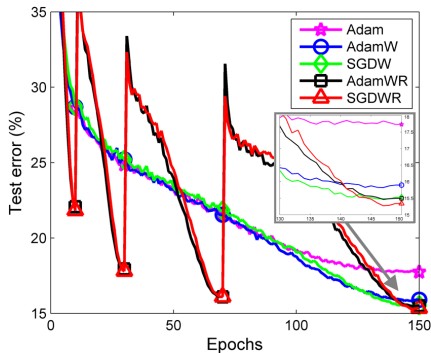

Figure 4: Top-1 test error on CIFAR-10 (left) and Top-5 test error on ImageNet32x32 (right). For a better resolution and with training loss curves, see SuppFigure 5 and SuppFigure 6 in the supplementary material.

### 4.4 AdamWR with Warm Restarts for Better Anytime Performance

In order to improve anytime performance of SGDW and AdamW we extended them with warm restarts of (Loshchilov & Hutter, 2016) to obtain SGDWR and AdamWR, respectively (see section B.2 in the Appendix). As Figure 4 shows, AdamWR greatly sped up AdamW on CIFAR-10 and ImageNet32x32, up to a factor of 10 (see the results at the first restart). For the default learning rate of 0.001, *AdamW achieved 15% relative improvement in test errors compared to Adam both on CIFAR-10* (also see Figure 3) *and ImageNet32x32* (also see SuppFigure 5). *AdamWR achieved the same improved results but with a much better anytime performance.* These improvements closed most of the gap between Adam and SGDWR on CIFAR-10 and yielded comparable performance on ImageNet32x32.

### 4.5 Use of AdamW on other datasets and architectures

Several other research groups have already successfully applied AdamW in citable works. For example, Wang et al. (2018) used AdamW to train a novel architecture for face detection on the standard WIDER FACE dataset (Yang et al., 2016), obtaining almost 10x faster predictions than the previous state of the art algorithms while achieving comparable performance. Völker et al. (2018) employed AdamW with cosine annealing to train convolutional neural networks to classify and characterize error-related brain signals measured from intracranial electroencephalography (EEG) recordings. While their paper does not provide a comparison to Adam, they kindly provided us with a direct comparison of the two on their best-performing problem-specific network architecture Deep4Net and a variant of ResNet. AdamW with the same hyperparameter setting as Adam yielded higher test set accuracy on Deep4Net (73.68% versus 71.37%) and statistically significantly higher test set accuracy on ResNet (72.04% versus 61.34%). Radford et al. (2018) employed AdamW to train Transformer (Vaswani et al., 2017) architectures to obtain new state-of-the-art results on a wide range of benchmarks for natural language understanding. Zhang et al. (2018) compared $L_2$ regularization vs. weight decay for SGD, Adam and the Kronecker-Factored Approximate Curvature (K-FAC) optimizer (Martens & Grosse, 2015) on the CIFAR datasets with ResNet and VGG architectures, reporting that decoupled weight decay consistently outperformed $L_2$ regularization in cases where they differ.

## 5 Conclusion and Future Work

Following suggestions that adaptive gradient methods such as Adam might lead to worse generalization than SGD with momentum (Wilson et al., 2017), we identified and exposed the inequivalence of $L_2$ regularization and weight decay for Adam. We empirically showed that our version of Adam with decoupled weight decay yields substantially better generalization performance than the common implementation of Adam with $L_2$ regularization. We also proposed to use warm restarts for Adam to improve its anytime performance.

Our results obtained on image classification datasets must be verified on a wider range of tasks, especially ones where the use of regularization is expected to be important. It would be interesting to integrate our findings on weight decay into other methods which attempt to improve Adam, e.g, normalized direction-preserving Adam (Zhang et al., 2017). While we focused our experimental analysis on Adam, we believe that similar results also hold for other adaptive gradient methods, such as AdaGrad (Duchi et al., 2011) and AMSGrad (Reddi et al., 2018).

## 6 ACKNOWLEDGMENTS

This work was supported by the European Research Council (ERC) under the European Union's Horizon 2020 research and innovation programme under grant no. 716721, by the German Research Foundation (DFG), under the BrainLinksBrainTools Cluster of Excellence (grant number EXC 1086) and through grant no. INST 37/935-1 FUGG, and by the German state of Baden-Württemberg through bwHPC. We thank Patryk Chrabaszcz for helping running experiments with ImageNet32x32. We thank Matthias Feurer and Robin Schirrmeister for providing valuable feedback on this paper in several iterations. We thank Martin Völker, Robin Schirrmeister, and Tonio Ball for providing us with a comparison of AdamW and Adam on their EEG data.

Finally, we thank the following members of the deep learning community for implementing decoupled weight decay in various deep learning libraries:

- Jingwei Zhang, Lei Tai, Robin Schirrmeister, and Kashif Rasul for their implementations in PyTorch (see `https://github.com/pytorch/pytorch/pull/4429`)
- Phil Jund for his implementation in TensorFlow described at `https://www.tensorflow.org/api_docs/python/tf/contrib/opt/DecoupledWeightDecayExtension`
- Sylvain Gugger, Anand Saha, Jeremy Howard and other members of fast.ai for their implementation available at `https://github.com/sgugger/Adam-experiments`
- Guillaume Lambard for his implementation in Keras available at `https://github.com/GLambard/AdamW_Keras`
- Yagami Lin for his implementation in Caffe available at `https://github.com/Yagami123/Caffe-AdamW-AdamWR`

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

# Appendix

## A  FORMAL ANALYSIS OF WEIGHT DECAY VS L$_2$ REGULARIZATION

**Proof of Proposition 1**
The proof for this well-known fact is straight-forward. SGD without weight decay has the following iterates on $f_t^{\text{reg}}(\boldsymbol{\theta}) = f_t(\boldsymbol{\theta}) + \frac{\lambda'}{2} \|\boldsymbol{\theta}\|_2^2$:

$$\boldsymbol{\theta}_{t+1} \leftarrow \boldsymbol{\theta}_t - \alpha\nabla f_t^{\text{reg}}(\boldsymbol{\theta}_t) = \boldsymbol{\theta}_t - \alpha\nabla f_t(\boldsymbol{\theta}_t) - \alpha\lambda'\boldsymbol{\theta}_t. \tag{5}$$

SGD with weight decay has the following iterates on $f_t(\boldsymbol{\theta})$:

$$\boldsymbol{\theta}_{t+1} \leftarrow (1-\lambda)\boldsymbol{\theta}_t - \alpha\nabla f_t(\boldsymbol{\theta}_t). \tag{6}$$

These iterates are identical since $\lambda' = \frac{\lambda}{\alpha}$. $\qquad\square$

**Proof of Proposition 2**
Similarly to the Proof of Proposition 1, the iterates of $O$ without weight decay on $f_t^{\text{reg}}(\boldsymbol{\theta}) = f_t(\boldsymbol{\theta}) + \frac{1}{2}\lambda'\|\boldsymbol{\theta}\|_2^2$ and $O$ with weight decay $\lambda$ on $f_t$ are, respectively:

$$\boldsymbol{\theta}_{t+1} \quad \leftarrow \quad \boldsymbol{\theta}_t - \alpha\lambda'\mathbf{M}_t\boldsymbol{\theta}_t - \alpha\mathbf{M}_t\nabla f_t(\boldsymbol{\theta}_t). \tag{7}$$

$$\boldsymbol{\theta}_{t+1} \quad \leftarrow \quad (1-\lambda)\boldsymbol{\theta}_t - \alpha\mathbf{M}_t\nabla f_t(\boldsymbol{\theta}_t). \tag{8}$$

The equality of these iterates for all $\boldsymbol{\theta}_t$ would imply $\lambda\boldsymbol{\theta}_t = \alpha\lambda'\mathbf{M}_t\boldsymbol{\theta}_t$. This can only hold for all $\boldsymbol{\theta}_t$ if $\mathbf{M}_t = k\mathbf{I}$, with $k \in \mathbb{R}$, which is not the case for $O$. Therefore, no L$_2$ regularizer $\lambda'\|\boldsymbol{\theta}\|_2^2$ exists that makes the iterates equivalent. $\qquad\square$

**Proof of Proposition 3**
$O$ without weight decay has the following iterates on $f_t^{\text{sreg}}(\boldsymbol{\theta}) = f_t(\boldsymbol{\theta}) + \frac{\lambda'}{2}\|\boldsymbol{\theta} \odot \sqrt{\boldsymbol{s}}\|_2^2$:

$$\boldsymbol{\theta}_{t+1} \quad \leftarrow \quad \boldsymbol{\theta}_t - \alpha\nabla f_t^{\text{sreg}}(\boldsymbol{\theta}_t)/\boldsymbol{s} \tag{9}$$

$$= \quad \boldsymbol{\theta}_t - \alpha\nabla f_t(\boldsymbol{\theta}_t)/\boldsymbol{s} - \alpha\lambda'\boldsymbol{\theta}_t \odot \boldsymbol{s}/\boldsymbol{s} \tag{10}$$

$$= \quad \boldsymbol{\theta}_t - \alpha\nabla f_t(\boldsymbol{\theta}_t)/\boldsymbol{s} - \alpha\lambda'\boldsymbol{\theta}_t, \tag{11}$$

where the division by $\boldsymbol{s}$ is element-wise. $O$ with weight decay has the following iterates on $f_t(\boldsymbol{\theta})$:

$$\boldsymbol{\theta}_{t+1} \quad \leftarrow \quad (1-\lambda)\boldsymbol{\theta}_t - \alpha\nabla f(\boldsymbol{\theta}_t)/\boldsymbol{s} \tag{12}$$

$$= \quad \boldsymbol{\theta}_t - \alpha\nabla f(\boldsymbol{\theta}_t)/\boldsymbol{s} - \lambda\boldsymbol{\theta}_t, \tag{13}$$

These iterates are identical since $\lambda' = \frac{\lambda}{\alpha}$. $\qquad\square$

## B  ADDITIONAL PRACTICAL IMPROVEMENTS OF ADAM

Having discussed decoupled weight decay for improving Adam's generalization, in this section we introduce two additional components to improve Adam's performance in practice.

### B.1  NORMALIZED WEIGHT DECAY

Our preliminary experiments showed that different weight decay factors are optimal for different computational budgets (defined in terms of the number of batch passes). Relatedly, Li et al. (2017) demonstrated that a smaller batch size (for the same total number of epochs) leads to the shrinking effect of weight decay being more pronounced. Here, we propose to reduce this dependence by normalizing the values of weight decay. Specifically, we replace the hyperparameter $\lambda$ by a new (more robust) normalized weight decay hyperparameter $\lambda_{norm}$, and use this to set $\lambda$ as $\lambda = \lambda_{norm}\sqrt{\frac{b}{BT}}$, where $b$ is the batch size, $B$ is the total number of training points and $T$ is the total number of epochs.[2] Thus, $\lambda_{norm}$ can be interpreted as the weight decay used if only one batch pass is allowed. We emphasize that our choice of normalization is merely one possibility informed by few experiments; a more lasting conclusion we draw is that using *some* normalization can substantially improve results.

---

[2] In the context of our AdamWR variant discussed in Section B.2, $T$ is the total number of epochs in the current restart.

## B.2 Adam with Cosine Annealing and Warm Restarts

We now apply cosine annealing and warm restarts to Adam, following the recent work of Loshchilov & Hutter (2016). There, the authors proposed Stochastic Gradient Descent with Warm Restarts (SGDR) to improve anytime performance of SGD by quickly cooling down the learning rate according to a cosine schedule and periodically increasing it. SGDR has been successfully adopted to lead to new state-of-the-art results for popular image classification benchmarks (Huang et al., 2017; Gastaldi, 2017; Zoph et al., 2017), and we therefore tried extending it to Adam. However, while our initial version of Adam with warm restarts had better anytime performance than Adam, it was not competitive with SGD with warm restarts, precisely because $L_2$ regularization was not working as well as in SGD. Now, having fixed this issue by means of the original weight decay regularization (Section 2) and also having introduced normalized weight decay (Section B.1), the original work on cosine annealing and warm restarts by Loshchilov & Hutter (2016) directly carries over to Adam.

In the interest of keeping the presentation self-contained, we briefly describe how SGDR schedules the change of the effective learning rate in order to accelerate the training of DNNs. Here, we decouple the initial learning rate $\alpha$ and its multiplier $\eta_t$ used to obtain the actual learning rate at iteration $t$ (see, e.g., line 8 in Algorithm 1). In SGDR, we simulate a new warm-started run/restart of SGD once $T_i$ epochs are performed, where $i$ is the index of the run. Importantly, the restarts are not performed from scratch but emulated by increasing $\eta_t$ while the old value of $\boldsymbol{\theta}_t$ is used as an initial solution. The amount by which $\eta_t$ is increased controls to which extent the previously acquired information (e.g., momentum) is used. Within the $i$-th run, the value of $\eta_t$ decays according to a cosine annealing (Loshchilov & Hutter, 2016) learning rate for each batch as follows:

$$\eta_t = \eta_{min}^{(i)} + 0.5(\eta_{max}^{(i)} - \eta_{min}^{(i)})(1 + \cos(\pi T_{cur}/T_i)),\qquad(14)$$

where $\eta_{min}^{(i)}$ and $\eta_{max}^{(i)}$ are ranges for the multiplier and $T_{cur}$ accounts for how many epochs have been performed since the last restart. $T_{cur}$ is updated at each batch iteration $t$ and is thus not constrained to integer values. Adjusting (e.g., decreasing) $\eta_{min}^{(i)}$ and $\eta_{max}^{(i)}$ at every $i$-th restart (see also Smith (2016)) could potentially improve performance, but we do not consider that option here because it would involve additional hyperparameters. For $\eta_{max}^{(i)} = 1$ and $\eta_{min}^{(i)} = 0$, one can simplify Eq. (14) to

$$\eta_t = 0.5 + 0.5\cos(\pi T_{cur}/T_i).\qquad(15)$$

In order to achieve good anytime performance, one can start with an initially small $T_i$ (e.g., from 1% to 10% of the expected total budget) and multiply it by a factor of $T_{mult}$ (e.g., $T_{mult} = 2$) at every restart. The $(i + 1)$-th restart is triggered when $T_{cur} = T_i$ by setting $T_{cur}$ to 0. An example setting of the schedule multiplier is given in C.

Our proposed **AdamWR** algorithm represents AdamW (see Algorithm 2) with $\eta_t$ following Eq. (15) and $\lambda$ computed at each iteration using normalized weight decay described in the previous section. We note that normalized weight decay allowed us to use a constant parameter setting across short and long runs performed within AdamWR and SGDWR (SGDW with warm restarts).

## C An Example Setting of the Schedule Multiplier

An example schedule of the schedule multiplier $\eta_t$ is given in SuppFigure 1 for $T_{i=0} = 100$ and $T_{mult} = 2$. After the initial 100 epochs the learning rate will reach 0 because $\eta_{t=100} = 0$. Then, since $T_{cur} = T_{i=0}$, we restart by resetting $T_{cur} = 0$, causing the multiplier $\eta_t$ to be reset to 1 due to Eq. (15). This multiplier will then decrease again from 1 to 0, but now over the course of 200 epochs because $T_{i=1} = T_{i=0}T_{mult} = 200$. Solutions obtained right before the restarts, when $\eta_t = 0$ (e.g., at epoch indexes 100, 300, 700 and 1500 as shown in SuppFigure 1) are recommended by the optimizer as the solutions, with more recent solutions prioritized.

## D Additional Results

We investigated whether the use of much longer runs (1800 epochs) of "standard Adam" (Adam with $L_2$ regularization and a fixed learning rate) makes the use of cosine annealing unnecessary.

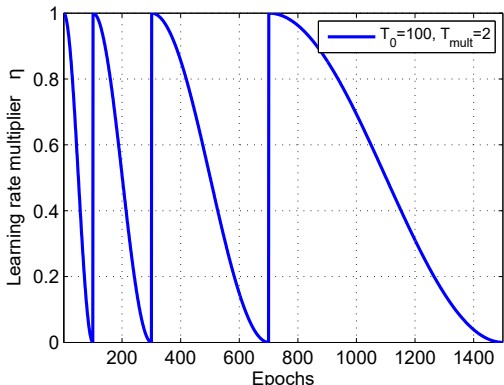

SuppFigure 1: An example schedule of the learning rate multiplier as a function of epoch index. The first run is scheduled to converge at epoch $T_{i=0} = 100$, then the budget for the next run is doubled as $T_{i=1} = T_{i=0}T_{mult} = 200$, etc.

SuppFigure 2 shows the results of standard Adam for a 4 by 4 logarithmic grid of hyperparameter settings (the coarseness of the grid is due to the high computational expense of runs for 1800 epochs). Even after taking the low resolution of the grid into account, the results appear to be at best comparable to the ones obtained with AdamW with 18 times less epochs and a smaller network (see SuppFigure 3, top row, middle). These results are not very surprising given Figure 2 in the main paper (which demonstrates the effectiveness of AdamW) and SuppFigure 1 (which demonstrates the necessity to use some learning rate schedule such as cosine annealing).

Our experimental results with Adam and SGD suggested that the total runtime in terms of the number of epochs affect the basin of optimal hyperparameters (see SuppFigure 3). More specifically, the greater the total number of epochs the smaller the values of the weight decay should be. SuppFigure 4 shows that our remedy for this problem, the normalized weight decay defined in Eq. (15), simplifies hyperparameter selection because the optimal values observed for short runs are similar to the ones for much longer runs. We used our initial experiments on CIFAR-10 to suggest the square root normalization we proposed in Eq. (15) and double-checked that this is not a coincidence on the ImageNet32x32 dataset (Chrabaszcz et al., 2017), a downsampled version of the original ImageNet dataset with 1.2 million $32 \times 32$ pixels images, where an epoch is 24 times longer than on CIFAR-10. This experiment also supported the square root scaling: the best values of the normalized weight decay observed on CIFAR-10 represented nearly optimal values for ImageNet32x32 (see SuppFigure 3). In contrast, had we used the same raw weight decay values $\lambda$ for ImageNet32x32 as for CIFAR-10 and for the same number of epochs, *without the proposed normalization, $\lambda$ would have been roughly 5 times too large for ImageNet32x32, leading to much worse performance*. The optimal normalized weight decay values were also very similar (e.g., $\lambda_{norm} = 0.025$ and $\lambda_{norm} = 0.05$) across SGDW and AdamW.

SuppFigure 4 is the equivalent of Figure 3 in the main paper, but for ImageNet32x32 instead of for CIFAR-10. The qualitative results are identical: weight decay leads to better training loss (cross-entropy) than $L_2$ regularization, and to an even greater improvement of test error.

SuppFigure 5 and SuppFigure 6 are the equivalents of Figure 4 in the main paper but supplemented with training loss curves in its bottom row. The results show that Adam and its variants with decoupled weight decay converge faster (in terms of training loss) on CIFAR-10 than the corresponding SGD variants (the difference for ImageNet32x32 is small). As is discussed in the main paper, when the same values of training loss are considered, AdamW demonstrates better values of test error than Adam. Interestingly, SuppFigure 5 and SuppFigure 6 show that restart variants AdamWR and SGDWR also demonstrate better generalization than AdamW and SGDW, respectively.

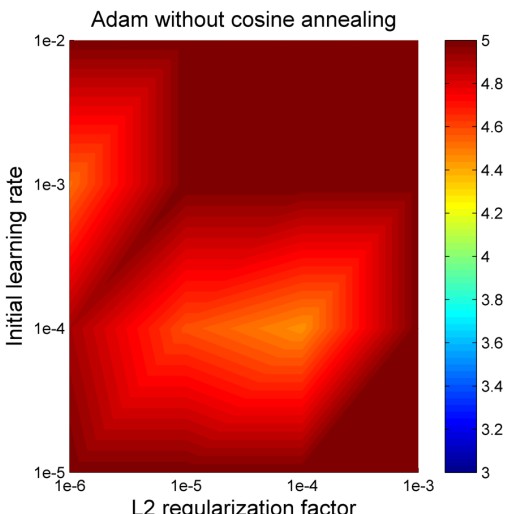

SuppFigure 2: Performance of "standard Adam": Adam with $L_2$ regularization and a fixed learning rate. We show the final test error of a 26 2x96d ResNet on CIFAR-10 after 1800 epochs of the original Adam for different settings of learning rate and weight decay used for $L_2$ regularization.

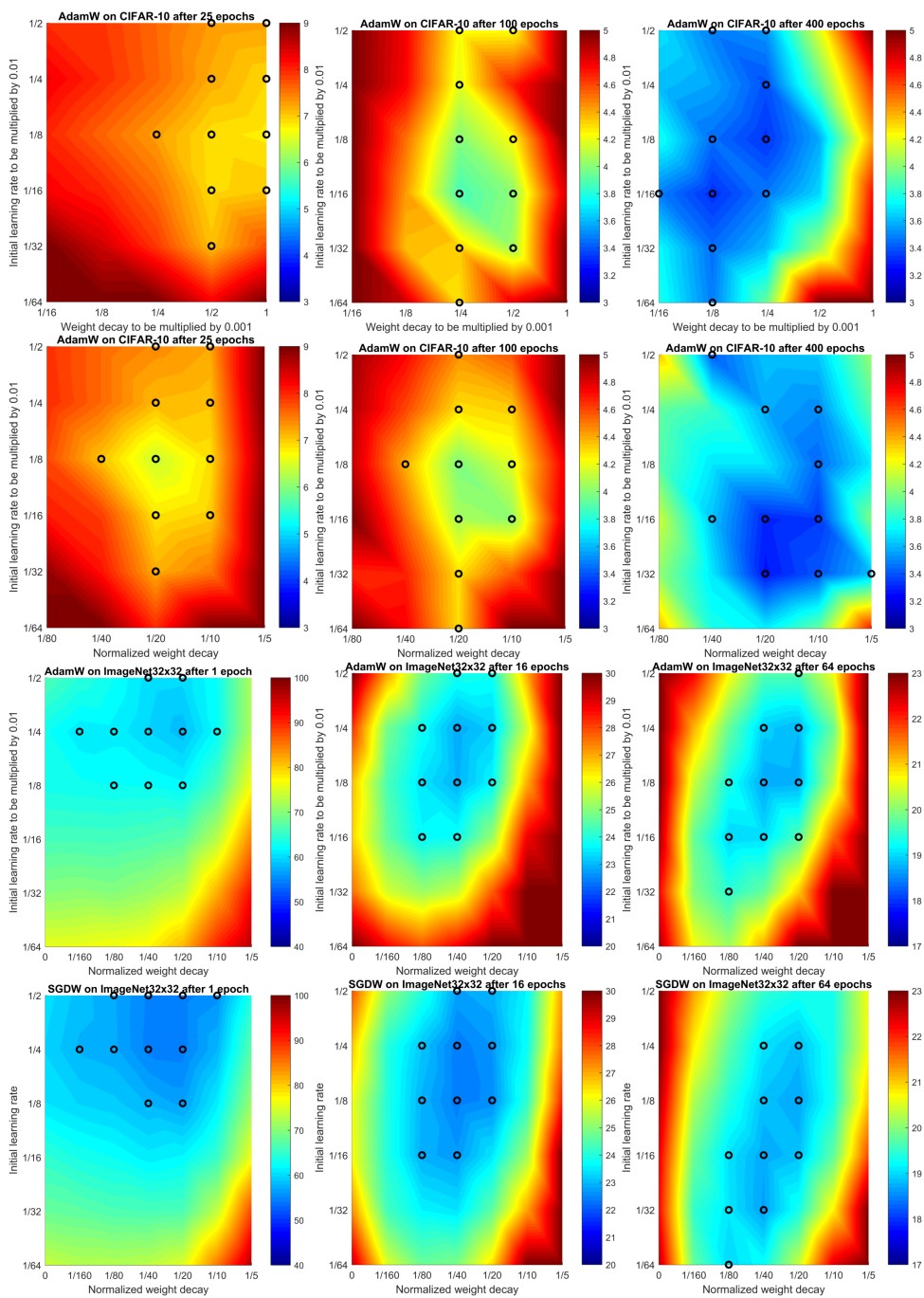

SuppFigure 3: Effect of normalized weight decay. We show the final test Top-1 error on CIFAR-10 (first two rows for AdamW without and with normalized weight decay) and Top-5 error on ImageNet32x32 (last two rows for AdamW and SGDW, both with normalized weight decay) of a 26 2x64d ResNet after different numbers of epochs (see columns). While the optimal settings of the raw weight decay change significantly for different runtime budgets (see the first row), the values of the normalized weight decay remain very similar for different budgets (see the second row) and different datasets (here, CIFAR-10 and ImageNet32x32), and even across AdamW and SGDW.

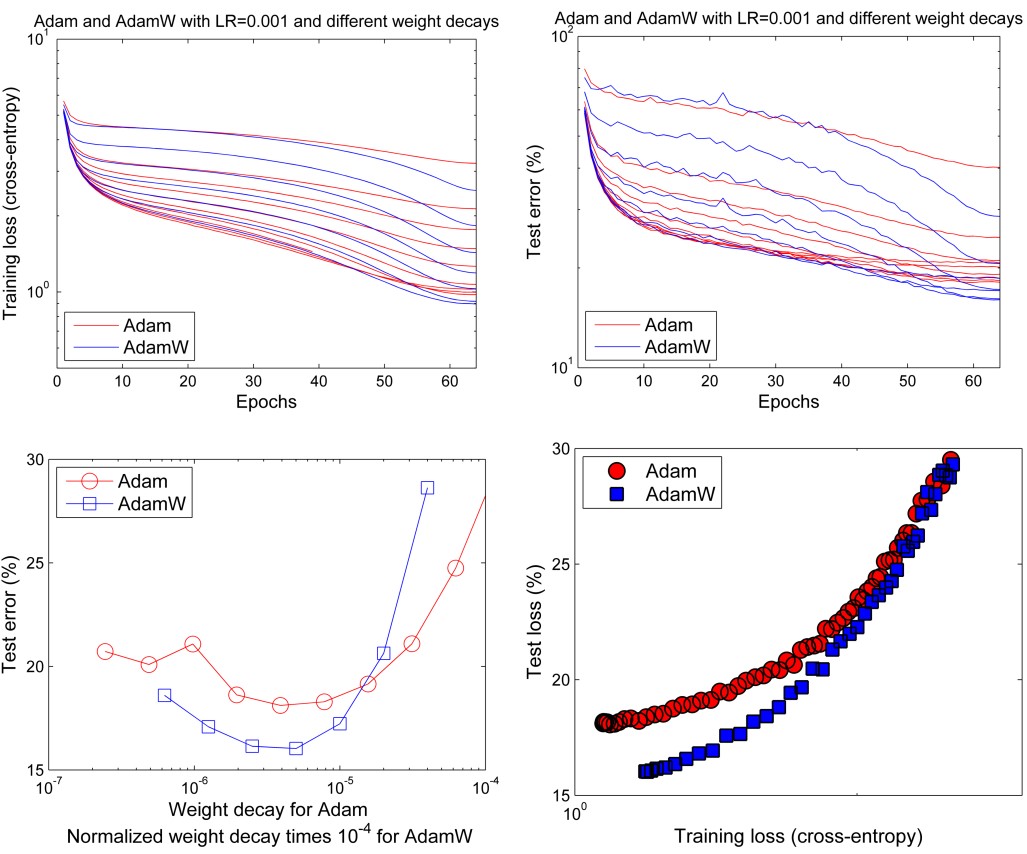

SuppFigure 4: Learning curves (top row) and generalization results (Top-5 errors in bottom row) obtained by a 26 2x96d ResNet trained with Adam and AdamW on ImageNet32x32.

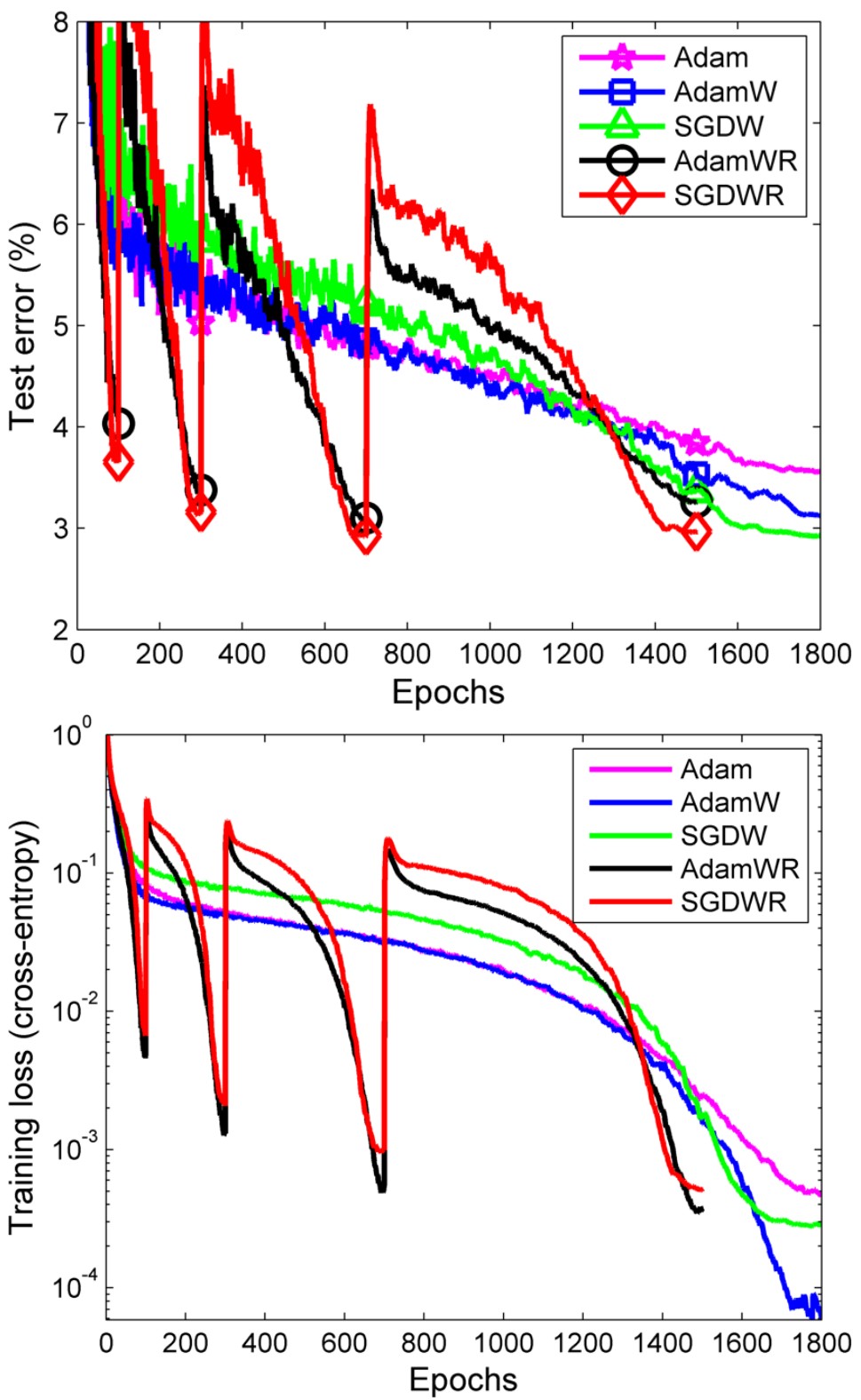

SuppFigure 5: Test error curves (top row) and training loss curves (bottom row) for CIFAR-10.

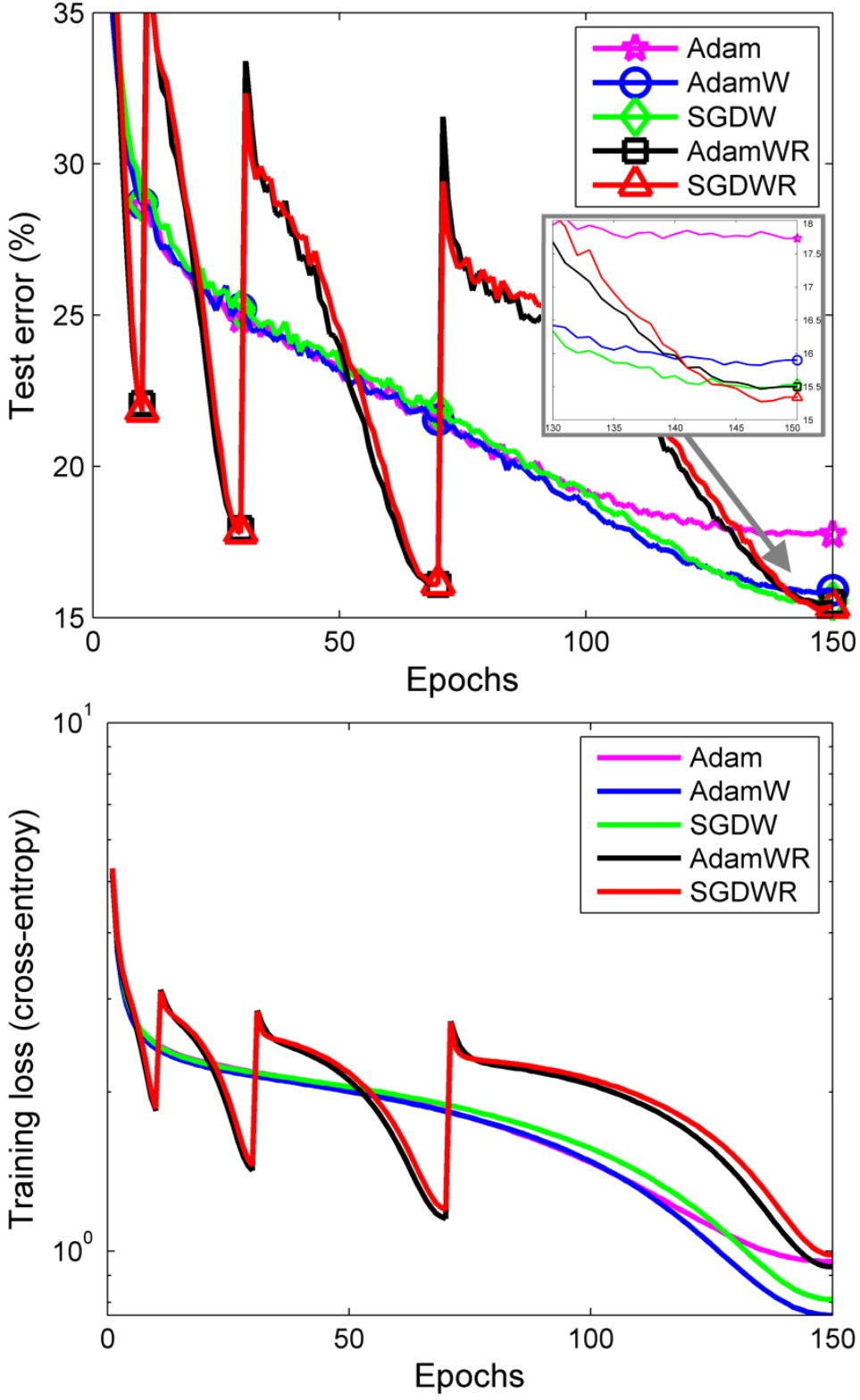

SuppFigure 6: Test error curves (top row) and training loss curves (bottom row) for ImageNet32x32.

