# OpenReview forum: "Decoupled Weight Decay Regularization"
_ICLR.cc/2019/Conference_

### Official Review · AnonReviewer2 · 2018-10-29

**Rating:** 5
**Confidence:** 4

**Review:**

This paper first identifies an inequivalence between L2 regularization and the original weight decay in adaptive stochastic gradient methods, e.g., the Adam method, and then proposes two decoupled variants, SGDW and AdamW, respective. The authors also cited a recent work to provide a justification of their proposed update rules from the perspective of Bayesian filtering. To demonstrate the effectiveness of both methods, experiments on CIFAR10 and ImageNet32x32 are conducted to compare with the original methods. Results show that the proposed methods consistently lead to faster convergence. Overall the paper is well written and easy to follow, with enough details describing the experimental settings.

First of all I appreciate the authors pointing out that weight decay is not equal to L2 regularization in general. This is evident once the original definition of weight decay is given. The main motivation comes from the argument that instead of using L2 regularization, weight decay should be used in adaptive gradient methods. The Bayesian filtering interpretation helps to justify the proposed method. But it is not clear to me why the hyperparameters w and \alpha are decoupled in the proposed methods? For example, in Line 6 of Alg. 1, g_t is a function of w, and later in Line 8, g_t is coupled with \alpha which naturally introduces a term w \alpha into m_t. So both w and \alpha are still coupled together in the proposed algorithm. If this is the case why the authors still call w and \alpha decoupled?

To me the most interesting result is Proposition 3 where the authors show that weight decay actually corresponds to preconditioned L2 regularization. This helps to explain what's the algorithmic difference between these two methods in adaptive gradient methods, and provides an intuitive insight on why weight decay may lead to better results compared with the vanilla L2 regularization.

Experiments on image recognition tasks basically confirm the authors' claims. However, as the authors have already pointed out, it is better to have more thorough experiments on other kinds of tasks, e.g., in text classification, etc. If the improvement does come from the difference between weight decay vs L2, then I would also expect the same improvement on other tasks. It would be great to see more experimental results on other tasks to have a better understanding of this problem. So far it is not clear whether the same improvement holds in general or not.

---

> ### Author Response · Authors · 2018-11-26
> **Response**
>
> Thank you for the positive and detailed review and your questions and comments. We reply to them below.
>
>                                                                                        ***
>
> “It would be great to see more experimental results on other tasks to have a better understanding of this problem.”
>
> Please see Section 4.5 where we now mention additional applications of our decoupled weight decay.
>
>                                                                                        ***
>
> “But it is not clear to me why the hyperparameters w and \alpha are decoupled in the proposed methods? For example, in Line 6 of Alg. 1, g_t is a function of w, and later in Line 8, g_t is coupled with \alpha which naturally introduces a term w \alpha into m_t. So both w and \alpha are still coupled together in the proposed algorithm. If this is the case why the authors still call w and \alpha decoupled?”
>
> We believe this question is due to a point of confusion: please note that in Algorithm 1 we use two different colors to define the one difference between SGD and SGDW: the text with the purple background in Line 6 is only active for SGD, and the part with the green background in Line 9 is only active for SGDW.
> To reply in detail, please note that we modified our notation as proposed by AnonReviewer3 to denote weights as \theta (in line with the original Adam paper) and the weight decay hyperparameter as \lambda. In order to avoid possible confusions, we first repeat your question with the new notation applied:
>
> “But it is not clear to me why the hyperparameters w and \alpha are decoupled in the proposed methods? For example, in Line 6 of Alg. 1, g_t is a function of \lambda, and later in Line 8, g_t is coupled with \alpha which naturally introduces a term \lambda \alpha into m_t. So both \lambda and \alpha are still coupled together in the proposed algorithm. If this is the case why the authors still call \lambda and \alpha decoupled?”
>
> You described the original SGD with L2 regularization when you mention “in Line 6 of Alg. 1, g_t is a function of \lambda, and later in Line 8, g_t is coupled with \alpha which naturally introduces a term \lambda \alpha into m_t”. It is true that this means that *in the original SGD* \alpha and \lambda are coupled. In contrast, SGDW does not have \lambda in Line 6 but has it in Line 9, where \lambda and \alpha are decoupled. Please note that Algorithm 1 uses two different colors to define specifics of SGD and SGDW when regularization is applied. We hypothesize that this confusion might have been caused by a black&white printout (in which the original colors were not easy to tell apart) and to avoid this confusion in the future we changed the colors to also be clearly distinguishable when printed in black&white.
>
>
> Thanks again for your review! We would kindly ask you to please consider updating your rating if this reply clarified your concerns, in particular about decoupling.

---

### Official Review · AnonReviewer3 · 2018-11-01
**I find the justification for decoupled weight decay a little unconvincing, but the empirical results are solid**

**Rating:** 7
**Confidence:** 4

**Review:**

This review has been somewhat challenging to complete. As the authors write, this work has already been impactful and motivated a great deal of further research. The empirical evaluation is convincing and the results have been reproduced and further studied by others. A moderate amount of space in the paper (Section 3, Section 4.5) is used to refer to work motivated by the paper itself. While I do not take issue with this I believe it should be considered for the final decision (in the sense that disentangling the contributions of the authors and related work becomes tricky). With this said, I continue with my review.

Paper summary: The authors observe that L2 regularization is not effective when using the Adam optimizer. By replacing L2 regularization with decoupled weight decay the authors are able to close the generalization gap between SGD and Adam and make Adam more robust to hyperparameter settings. The empirical evaluation is comprehensive and convincing.

Detailed comments:

1) The authors emphasize the fact that L2 regularization and weight decay are not the same for different optimizers and claim that this goes against the belief of some practitioners. In my experience, most practitioners would not be surprised by this observation itself. The second observation made by the authors, that L2 regularization is not effective in Adam, is the more interesting (and perhaps surprising) observation.

2) I am not convinced of the importance of Proposition 3. In practice, adaptive methods will have a preconditioner which depends locally on the parameters. I understand the motivation from the previous paragraph but felt that the formal result added little.

3) Section 3 introduced the Bayesian filtering perspective of stochastic optimization. The authors share the observation of Aitchison, 2018 that decoupled weight decay can be recovered in this framework. My interpretation is that this observation is important _because_ of the empirical observations in this paper and does not necessarily provide theoretical support for the approach. However, the last paragraph of Section 3 seems to utilize the Bernstein-von Mises theorem to promote the idea that with large datasets the prior distribution is unimportant (and is ignored). I am not sure that I follow this argument. For example, this claim seems to be completely independent of the optimization algorithm used and moreover Propositions 1,2, and 3 are independent of the data distribution. I suspect that this confusion is due to a misunderstanding on my part and would appreciate clarification from the authors.

4) The empirical evaluation in this paper is very strong and these practical techniques have already been adopted by the community in addition to spurring novel research. The empirical observation broadly explores two directions: decoupled weight decay leads to separable hyperparameter search spaces (meaning optimization is less sensitive to hyperparameters), and decoupled weight decay gives improved generalization (and training performance). Both claims are explored throughly with strong evidence given for the improvement due to AdamW.

Overall, I find this paper to be presented well and with convincing empirical results. I feel that the theoretical justification for decoupling weight decay are a little weak, and believe that other work is moving towards better explanations then the ones presented in this paper [1,2,3]. Despite this, I believe that this paper should be accepted.


Minor comments:

- I find the notation in the paper confusing in general. x is used to denote weights, and w to denote hyperparameters (e.g. w' for L2 regularization scale and w for weight decay scale). I don't see why it wouldn't be preferable to use the more standard W for weights, x for inputs, and lambda for hparams.
- Figure 4: it is difficult to distinguish between Adam and SGDWR (especially left).



Clarity: The paper is well written and clear. I find the notation confusing in places, but is consistent throughout.

Originality: This paper presents original findings but occasionally relies on work motivated by itself to convince the reader of its importance. I do not think that this subtracts from the value of the work.

Significance: The work is clearly significant. Even without knowing that practitioners have adopted the techniques presented in this work, the paper clearly distinguishes itself with strong empirical results.

---

> ### Author Response · Authors · 2018-11-26
> **Response**
>
> Thank you for the positive and detailed review and your questions and comments. We reply to them below.
>
>                                                                                        ***
>
> “1) The authors emphasize the fact that L2 regularization and weight decay are not the same for different optimizers and claim that this goes against the belief of some practitioners. In my experience, most practitioners would not be surprised by this observation itself. The second observation made by the authors, that L2 regularization is not effective in Adam, is the more interesting (and perhaps surprising) observation.”
>
> In our experience, practitioners often have the equivalence result for SGD in mind and other cases such as Adam tend to remain unnoticed until they actively think about them. In reply to your comment, we’ve toned down the wording from
>
> “Contrary to a belief which seems popular among some practitioners, the two techniques are not equivalent. For SGD, they can be made equivalent by a reparameterization of the weight decay factor based on the learning rate; this is not the case for Adam.”
>
> to
>
> “The two techniques can be made equivalent for SGD by a reparameterization of the weight decay factor based on the learning rate; however, as is often overlooked, this is not the case for Adam.”
>
>                                                                                        ***
>
> “2) I am not convinced of the importance of Proposition 3. In practice, adaptive methods will have a preconditioner which depends locally on the parameters. I understand the motivation from the previous paragraph but felt that the formal result added little.”
>
> We agree that the relevance of Proposition 3 does not derive from its immediate applicability to practical adaptive gradient algorithms. Nevertheless, we still believe the proposition to be useful since (for this simple special case of a fixed preconditioner) it provides a precise equivalence between decoupled weight decay and standard L2 regularization with a scaled regularizer and thus provides an intuitive explanation for what decoupled weight decay does: parameters with a small preconditioner (which in practice would be caused by typically large gradients in this dimension) are regularized relatively more than they would be with L2 regularization; specifically, the regularization is proportional to the inverse of the root of the preconditioner.
> For adaptive gradient algorithms with changing preconditioner matrices (which includes all popular cases) there is no 1-to-1 equivalence to a fixed L2 regularization, but we can still use the intuition from the proposition to think about what loss function is being optimized in each step.
>
>                                                                                        ***
>
> "However, the last paragraph of Section 3 seems to utilize the Bernstein-von Mises theorem to promote the idea that with large datasets the prior distribution is unimportant (and is ignored). I am not sure that I follow this argument. For example, this claim seems to be completely independent of the optimization algorithm used and moreover Propositions 1,2, and 3 are independent of the data distribution. I suspect that this confusion is due to a misunderstanding on my part and would appreciate clarification from the authors."
>
> Thank you for this comment! After a closer analysis of the argument by Aitchison described in the last paragraph of Section 3, we are also less convinced about it: due to the equivalence of L2 regularization and weight decay in SGD settings (our Proposition 1), one should not expect them to scale differently with the dataset size, as the argument would suggest. In order to avoid possible confusion, we decided to remove that paragraph entirely. Thank you for raising concerns about it.
>
>                                                                                        ***
>
> “I find the notation in the paper confusing in general. x is used to denote weights, and w to denote hyperparameters (e.g. w' for L2 regularization scale and w for weight decay scale). I don't see why it wouldn't be preferable to use the more standard W for weights, x for inputs, and lambda for hparams”
>
> Thanks, this is a good point. In response, we’ve now modified our notation to denote weights as \theta (in line with the original Adam paper) and the weight decay hyperparameter as \lambda.
>
>                                                                                        ***
>
> “Figure 4: it is difficult to distinguish between Adam and SGDWR (especially left).”
>
> We added SuppFigure 5 and SuppFigure 6 which should improve readability of Figure 4.
>
>
> Thanks again for your review!

---

> > ### Comment · AnonReviewer3 · 2018-11-26
> > **Response to response**
> >
> > 1) This completely clears up my concern.
> >
> > 2) It seems that we largely share the same opinion here. After some more reflection, I think that this proposition does bring some good to the paper by attempting to formalize the relationship between L2 regularization and weight decay in adaptive methods and agree with your comments in that regard.
> >
> > 3) Thank you for the response and clarification. I am glad that this was brought into the discussion!
> >
> >
> > ----
> >
> > Notation comment: Thank you for making this change -- I think it is much clearer now. With a quick pass through, everything looks consistent.
> >
> > Readability: The new supp figures are easy to read. I guess there is no easy fix for Figure 4, but I still consider this a minor issue.
> >
> >
> > New minor issue:
> >
> > In Supp figure 4 explanation in appendix: "and to an even greater improvements of test error". Improvements should not be plural.
> >
> > -------
> >
> > To summarize, I think that the revised paper improves on many of the minor issues I had with the original paper. I am still a little unconvinced by the theoretical justification but I feel that the empirical results and some of the formal analysis makes up for this. I hope that this paper is accepted.

---

> > > ### Author Response · Authors · 2018-11-27
> > > **Re: Response to response**
> > >
> > > Thank you very much for your positive evaluation! We have fixed the typo and updated the paper.

---

### Official Review · AnonReviewer1 · 2018-11-02
**Good paper, several concerns**

**Rating:** 6
**Confidence:** 4

**Review:**

In this paper, the authors investigate a very simple but still very interesting idea of decoupling weight decay and gradient step. It is a well known problem that Adam optimization method leads to worse generalization and stronger overfitting than SGD with momentum on classification tasks despite its faster convergence. The authors tried to find a reason for such behavior. They noticed that while SGD with L2 regularization is equivalent to SGD with weight decay, it is not the case for adaptive methods, such as Adam. The main contributions include the following:
1.  Improvement of Adam method via decoupling weight decay and optimization step and using warm restarts. The authors thoroughly investigated the proposed idea on different learning rate schedules and different datasets. It would also be interesting to see results on architectures other than ResNet. In section 4.5 the authors claim that the proposed idea was used in different settings by many authors. So, I would recommend to elaborate on this section in the final version of the paper.
2.  Reducing sensitivity of SGD to weight decay parameter. The authors noticed that the optimal weight decay parameter depends on the number of training epochs, therefore they proposed a functional form of dependency between weight decay and the number of batch passes.

I also have the following concerns:
1. One of the main advantages of Adam is the speed of convergence. Does AdamW or AdamWR converge faster than the corresponding SGD method? Figure 4 is not quite representative since it contains an experiment with a very large number of training epochs.
2. While AdamWR delivers much better test accuracy than Adam, it is still slightly worse than SGDWR method.

I would also recommend to change scale of y-axis, Figure 4, right. Since 0.5% percent difference can be significant for state-of-the-art classification results.


Overall, the paper is written clearly and organized well. It contains a lot of experiments and proposes an explanation of the observed phenomena. While the idea is very simple, the experimental results show its efficiency.

---

> ### Author Response · Authors · 2018-11-26
> **Response**
>
> Thank you for the positive detailed review and your questions and comments. We reply to them below.
>
>                                                                                       ***
>
> “It would also be interesting to see results on architectures other than ResNet. In section 4.5 the authors claim that the proposed idea was used in different settings by many authors. So, I would recommend to elaborate on this section in the final version of the paper.”
>
> Please see Section 4.5 where we now mention additional applications of our decoupled weight decay and AdamW.
>
>                                                                                        ***
>
> “1. One of the main advantages of Adam is the speed of convergence. Does AdamW or AdamWR converge faster than the corresponding SGD method? Figure 4 is not quite representative since it contains an experiment with a very large number of training epochs.”
>
> To address this question, we added SuppFigure 5, SuppFigure 6 and the following text in the supplementary material:
>
> “SuppFigure 5 and SuppFigure 6 are the equivalents of Figure 4 in the main paper but supplemented with training loss curves in its bottom row. The results show that Adam and its variants with decoupled weight decay converge faster (in terms of training loss) on CIFAR-10 than the corresponding SGD variants (the difference for ImageNet32x32 is small). As is discussed in the main paper, when the same values of training loss are considered, AdamW demonstrates better values of test error than Adam. Interestingly, SuppFigure 5 and SuppFigure 6 show that restart variants AdamWR and SGDWR also demonstrate better generalization than AdamW and SGDW, respectively. ”
>
> While in the paper we noted that restarts help to obtain better anytime performance, we didn’t pay attention to the fact that they also show better test errors for the same levels of training errors; this observation was made while answering this question, thank you.
>
>                                                                                        ***
>
> “2. While AdamWR delivers much better test accuracy than Adam, it is still slightly worse than SGDWR method.”
>
>
> We agree that some difference is still present on CIFAR-10 while the two are almost indistinguishable on ImageNet32x32. The largest part of the difference between SGD and Adam was linked to weight decay and L2 regularization, but we believe that this is not the case anymore for SGDW and AdamW. We tend to believe that the issues often cited for “adaptive methods may converge to sharp local optima” are present and we hope that our findings on weight decay regularization will complement new methods which attempt to address these issues.
>
>                                                                                        ***
>
> “I would also recommend to change scale of y-axis, Figure 4, right. Since 0.5% percent difference can be significant for state-of-the-art classification results.”
>
> Thanks, we agree and modified Figure 4, right, in response to this question. The new version includes a subfigure which better shows the very last epochs.
>
>
> Thanks again for your review!

---

### Author Response · Authors · 2018-11-26
**Response to All Reviewers**

We thank all reviewers for their positive evaluation and their valuable comments. We've uploaded a revision to address the issues raised and individually reply to the reviewers' concerns. We kindly ask you to update your rating if our replies clarified your concerns.
Thank you again for your reviews!

---

### Meta-Review · Area_Chair1 · 2018-12-05
**a useful and influential finding**

**Confidence:** 5
**Recommendation:** Accept (Poster)

**Metareview:**

Evaluating this paper is somewhat awkward because it has already been through multiple reviewing cycles, and in the meantime, the trick has already become widely adopted and inspired interesting follow-up work. Much of the paper is devoted to reviewing this follow-up work. I think it's clearly time for this to be made part of the published literature, so I recommend acceptance. (And all reviewers are in agreement that the paper ought to be accepted.)

The paper proposes, in the context of Adam, to apply literal weight decay in place of L2 regularization. An impressively thorough set of experiments are given to demonstrate the improved generalization performance, as well as a decoupling of the hyperparameters.

Previous versions of the paper suffered from a lack of theoretical justification for the proposed method. Ordinarily, in such cases, one would worry that the improved results could be due to some sort of experimental confound. But AdamW has been validated by so many other groups on a range of domains that the improvement is well established. And other researchers have offered possible explanations for the improvement.